# Magnetic Resonance Imaging to Diagnose and Predict the Outcome of Diabetic Kidney Disease—Where Do We Stand?

**Menno Pruijm** [1,*], **Ibtisam Aslam** [2], **Bastien Milani** [3], **Wendy Brito** [1], **Michel Burnier** [1], **Nicholas M. Selby** [4] and **Jean-Paul Vallée** [2]

1 Service of Nephrology, University Hospital of Lausanne and University of Lausanne, 1011 Lausanne, Switzerland; wendy.brito@chuv.ch (W.B.); michel.burnier@chuv.ch (M.B.)
2 Division of Radiology, Geneva University Hospitals, University of Geneva, 1205 Geneva, Switzerland; ibtisam.aslam@unige.ch (I.A.); jean-paul.vallee@hcuge.ch (J.-P.V.)
3 Center for Biomedical Imaging (CIBM-CHUV), Centre Hospitalier Universitaire Vaudois (CHUV), 1011 Lausanne, Switzerland; bastien.milani@chuv.ch
4 Centre for Kidney Research and Innovation (CKRI), Division of Health Sciences and Graduate Entry Medicine, University of Nottingham, Nottingham DE22 3DT, UK; nicholas.selby@nottingham.ac.uk
* Correspondence: menno.pruijm@chuv.ch; Tel.: +41-213141131; Fax: +41-213141139

**Abstract:** Diabetic kidney disease (DKD) is a major public health problem and its incidence is rising. The disease course is unpredictable with classic biomarkers, and the search for new tools to predict adverse renal outcomes is ongoing. Renal magnetic resonance imaging (MRI) now enables the quantification of metabolic and microscopic properties of the kidneys such as single-kidney, cortical and medullary blood flow, and renal tissue oxygenation and fibrosis, without the use of contrast media. A rapidly increasing number of studies show that these techniques can identify early kidney damage in patients with DKD, and possibly predict renal outcome. This review provides an overview of the currently most frequently used techniques, a summary of the results of some recent studies, and our view on their potential applications, as well as the hurdles to be overcome for the integration of these techniques into the clinical care of patients with DKD.

**Keywords:** BOLD-MRI; phase–contrast MRI; T1 mapping; arterial spin labelling; CKD

## 1. Introduction

It is well known that the epidemic of diabetes mellitus (DM) is ongoing. A substantial number of patients with DM will develop diabetic kidney disease (DKD), defined as the presence of (micro) albuminuria and/or an estimated glomerular filtration rate (eGFR) < 60 mL/min/1.73 m$^2$ [1]. In recent surveys, 40–50% of patients with type 2 DM (T2DM) developed DKD [2]. Despite the development of drugs that retard the progression of DKD such as renin–angiotensin blockers and, more recently, sodium–glucose cotransporter-2 (SGLT2) inhibitors and finerenone, the number of diabetic patients with kidney failure is still increasing [3]. As such, DKD has been the leading cause of patients on dialysis in the US since 2006 [4], with a similar trend in many other Western countries.

The early identification of DKD patients is important in order to implement nephroprotective measures, but is difficult to achieve due to the low sensitivity of classical biomarkers to detect renal structural damage. Creatinine is the most widely used marker of glomerular function, but typically, considerable renal damage has already occurred before creatinine starts to rise above the normal range.

As for albuminuria, it has become clear that more than 30% of T2DM patients have a progressive decline in eGFR despite the absence of albuminuria [5].

In addition, the progression rate of DKD is highly heterogeneous: some patients show little or no eGFR decline over the years, whereas others rapidly progress to kidney failure [6].

Therefore, there is an unmet need for new, non-invasive tools to identify structural renal damage at an early stage. There is also an unmet need to predict outcomes in a more reliable way, which will allow better health care planning and timely preparation for kidney replacement therapy (KRT), if necessary.

Conventional magnetic resonance imaging (MRI) of the kidneys is typically used to characterise tumours, quantify kidney volume in patients with polycystic kidney diseases, or to diagnose renal artery stenosis. At first sight, MRI is not an obvious candidate to visualise the renal microstructure or to predict renal outcome, as the obtained resolution is not high enough to depict glomeruli or other renal microstructures.

Nevertheless, the last decades have witnessed the development of several new MRI techniques enabling the quantification of metabolic and microscopic properties of the kidneys such as single-kidney, cortical and medullary blood flow, renal tissue oxygenation, and even fibrosis [7]. These parameters are not specific to DKD, but are of high interest, as all of them are involved in its pathophysiology. For example, biopsy studies in humans have shown that non-albuminuric DKD patients have more widespread interstitial fibrosis than those with albuminuria [8]. Moreover, diffuse intra-renal vascular rarefaction has been described in dissected kidneys from animals with DKD [9]. Vascular damage leads to renal hypoxia, proposed by many as the unifying mechanism for the development of DKD and other forms of CKD [10]. Animal studies using oxygen-sensing probes have indeed demonstrated that hypoxia is present in insulinopenic mice and precedes the onset of albuminuria and kidney damage [11].

These new MRI techniques—referred to by some as functional MRI—are not yet available in the clinic, and are only used in research settings. However, most of the specific sequences needed to measure the variables mentioned above can be installed without major hardware limitations on any modern MRI scanner. The number of studies that illustrate the correlations between MRI parameters, microstructure, blood flow, and the metabolic functions of the kidneys is rapidly increasing, and the results are promising. Therefore, many experts expect that some of these techniques will be introduced in clinical practice in the near future. The aims of this article are to briefly review the most promising techniques in a comprehensive way, to summarise the main results of recent studies (in our opinion), and to transmit our personal views on the potential use of renal MRI in DKD patients.

## 2. Overview of Available Renal MRI Techniques

Below we provide a short overview of the technical principles underlying each MRI technique. For those who wish to obtain more details and may consider integrating these techniques, we refer to recently published consensus-based recommendations [12–16]. These recommendations have been formulated by international experts as part of the COST action CA16103 PARENCHIMA (see https://www.renalmri.org (accessed on 14 March 2022)).

### 2.1. T1 and T2 Mapping

MRI is based on the interaction of proton nuclear spins and an external magnetic field called B0 after an excitation by different patterns of radiofrequency (RF) pulses. This spin perturbation generates an electromagnetic induction in receiver coils that can be registered to build an image. The spins relax back through two main mechanisms called T1 and T2 relaxations. The T1 relaxation, also called longitudinal relaxation, describes the return of the spins to their original position along B0. The T2 relaxation results from the energy dissipation between the spins and is characterised by an exponential decay of the electromagnetically induced signal. Repeating the acquisition with different timings of the RF pulse patterns enables the estimation of the physical T1 and T2 relaxation times for each voxel (i.e., the smallest part of the image), which yields the T1 and T2 maps.

Different protocols are available to acquire the images and build T1 or T2 maps, and we refer to the consensus paper for a more detailed description [14]. The $T_1$ values are influenced by the degree of oedema, hydration, and fibrosis. The T1 values are increased in inflammatory kidney diseases such as IgA nephropathy, and also increase in acute kidney

injury [17] or fibrosis [18]. The T2 values are also, if not more, sensitive to oedema and inflammation [19]. Closely linked to T2 is T2*, which is sensitive to local magnetic field heterogeneity as encountered by differences in deoxyhaemoglobin concentrations in tissue, as discussed in more detail under BOLD-MRI (Figure 1).

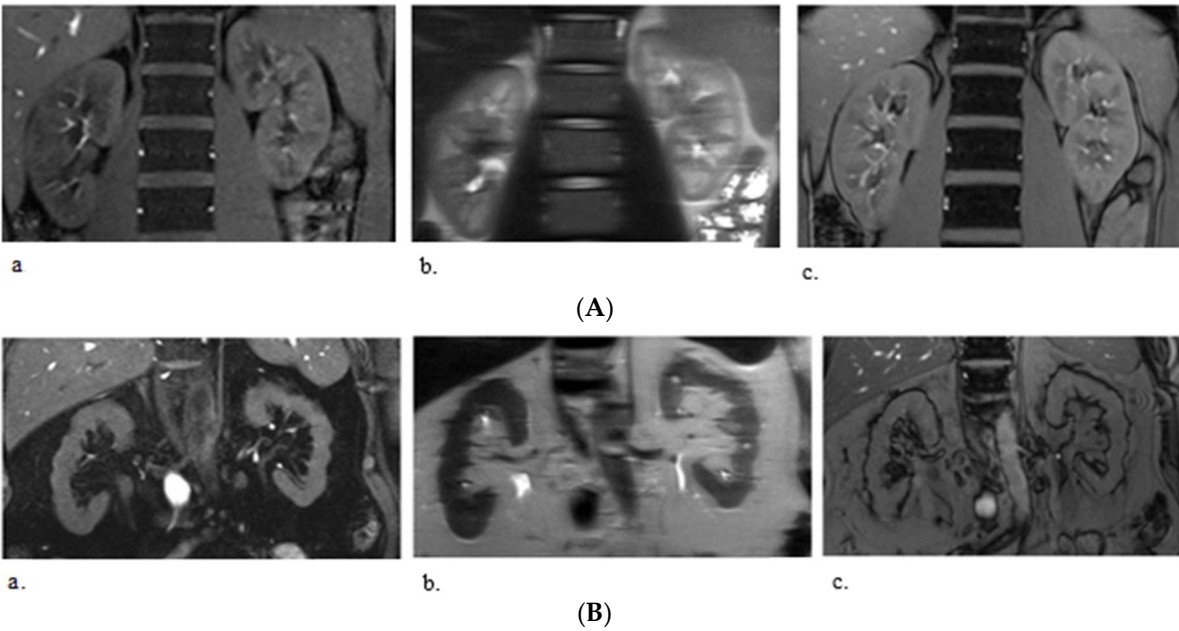

**Figure 1.** Examples of T1, T2, and T2* MRI in (**A**/**a**) 35-year-old, healthy man with an estimated GFR of 115 mL/min/1.73 m², and (**B**/**a**) 78-year-old man with DKD and an eGFR of 22 mL/min/1.73 m². In both subjects, different acquisitions were performed, with respectively (**a**) T1 FLASH, (**b**) T2 HASTE, and (**c**) T2* ponderation.

## 2.2. Phase–Contrast MRI

Phase–contrast (PC) MRI is a technique that measures blood flow separately in the right and left renal artery, without the need for contrast agents. This technique is based on phase shift differences of the spins of moving protons present in the blood in comparison to static protons in solid organs. It can be mathematically derived that the phase shift of each moving proton is proportional to its velocity [20]. This assumption is only true when the image is acquired in a segment of the renal artery with a constant, non-turbulent flow (so at a distance from any stenosis, in a straight segment of the artery; see Figure 2). Once the velocity is measured (in cm/sec), multiplying it by 60 and by the surface section of the artery (in cm²) provides the per-kidney renal blood flow (RBF) in ml/min. Hereafter, the renal vascular resistance can be calculated as the ratio between RBF and the mean arterial pressure. Renal PC-MRI has been validated in elegant phantom studies, but also against para-amino hippurate (PAH) clearance and hippuran nephrograms [20,21].

## 2.3. Arterial Spin Labelling

Arterial spin labelling (ASL) MRI is a subtraction technique that uses magnetically labelled water protons in the blood to measure cortical (and medullary) perfusion. Basically, two series of images are collected: labelled images and control images. By subtracting the two images, one obtains perfusion-weighted images in which the signal intensity is proportional to perfusion. Perfusion maps are obtained voxel by voxel, and the results can be expressed separately for the cortex and medulla. Cortical perfusion is in the range of 140–430 mL/100 g/min in healthy volunteers, whereas medullary perfusion is much lower, between 40 and 150 mL/100 g/min [22]. Validation is hampered by the lack of a gold standard technique to quantify intra-renal cortical and medullary perfusion in humans.

However, ASL-assessed micro-perfusion correlates well with microsphere techniques in animals [23], and phantoms have been recently developed to allow further validation [15].

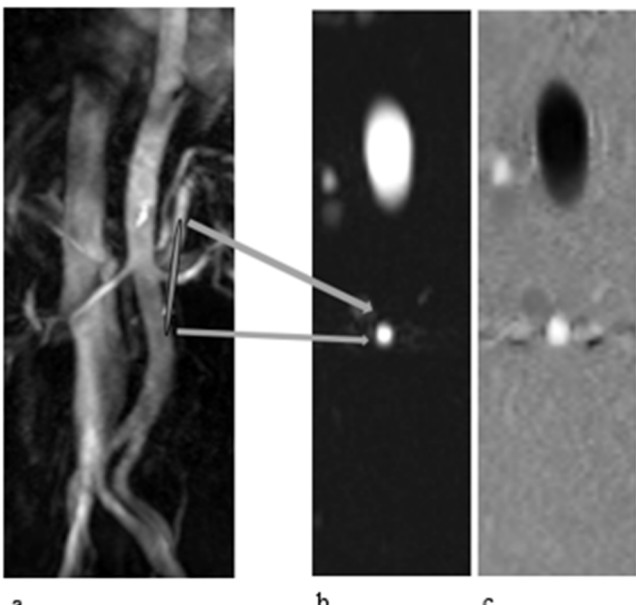

**Figure 2.** Example of phase–contrast MRI of the left renal artery in a 35-year-old healthy man. (**a**) Shown is a maximum-intensity projection of the entire aorta, used to localise the left renal artery (halfway along the bar). In order to obtain accurate velocities, an exact orthogonal plane of the vessel has to be selected; (**b**,**c**) show the resulting images of respectively the magnitude of the signal (**b**) and the velocities (**c**) of the left renal artery in a sagittal oblique plane. Post-processing allows quantification of the blood velocity in the vessel by placing ROIs in the lumen of the artery.

*2.4. BOLD-MRI*

Blood oxygenation level-dependent MRI (BOLD-MRI) is a technique that allows the estimation of renal tissue oxygenation. BOLD-MRI uses the paramagnetic properties of deoxyhaemoglobin (deoxyHb) to estimate renal tissue oxygenation. Oxygen is transported in the blood by haemoglobin (oxyHb). When oxygen is released, oxyHb becomes deoxyHb. Due to its paramagnetic properties, deoxyHb creates local magnetic susceptibility gradients in and around the blood vessels. This leads to a shortening of the relaxation times of T2* weighted images (see Figure 2). The higher the local deoxyHb, the shorter the T2* times of the tissue, and the faster the decline rate of R2* (=1/T2*, expressed as $s^{-1}$), which is usually the outcome variable of BOLD-MRI, although T2* can also be reported [24]. A high R2* value of a voxel corresponds to low local oxygenation, and vice versa. BOLD images can be acquired on 1.5 or 3T scans, but 3T is preferred. In general, three to five coronal slices are obtained, and with post-processing programs, R2* maps of each slice are built. The BOLD maps provide R2* values of each slice voxel per voxel (pixel per pixel).

BOLD-MRI is based on the assumption that blood deoxyHb levels correspond inversely with local tissular $pO_2$ levels. Small animal studies have indeed reported that there is an inverse and linear relationship between directly measured cortical and medullary $pO_2$ (with micro-probes) and the R2* values of the kidney [25]. However, the factors that modify the blood volume fraction of each voxel (oedema, for example) may also alter the R2* values without necessarily changing the oxygenation, so the results should be interpreted with caution [26].

*2.5. Diffusion MRI*

Diffusion-weighted imaging MRI (DWI-MRI) has emerged as a viable approach for assessing kidney microstructures, especially fibrosis. DWI is sensitive to water movement and uses diffusion gradients to establish imaging contrast in renal tissue and quantify the

motion of water in the tissue over time [27]. The motion of water in the presence of strong magnetic field gradients generates a signal attenuation used for DWI-MRI. The apparent diffusion coefficient (ADC) is a measurement of total water transport and microcirculation in the tissue. DWI determines ADC by acquiring the images with numerous diffusion-weighting factors called b-values [28].

DWI-MRI indirectly estimates the amount of renal fibrosis and/or cellular infiltration. As fibrotic tissue reduces the motion of the water molecules, a decreased ADC may suggest higher fibrosis. In healthy kidneys, water diffusion is larger in the cortex than in the medulla; however, in DKD patients and those with other forms of CKD, ADC may be lower in the cortex than in the medulla due to fibrosis. As a result, the difference between the cortical and medullary ADC (called ΔADC) can be negative in the case of advanced fibrosis (see Figure 3). In analogy with BOLD-MRI, physiological parameters other than fibrosis (for example, renal perfusion and hydration status) may also influence the DWI signal and ADC values [29]. Therefore, DWI-MRI should be performed under standardised hydration conditions.

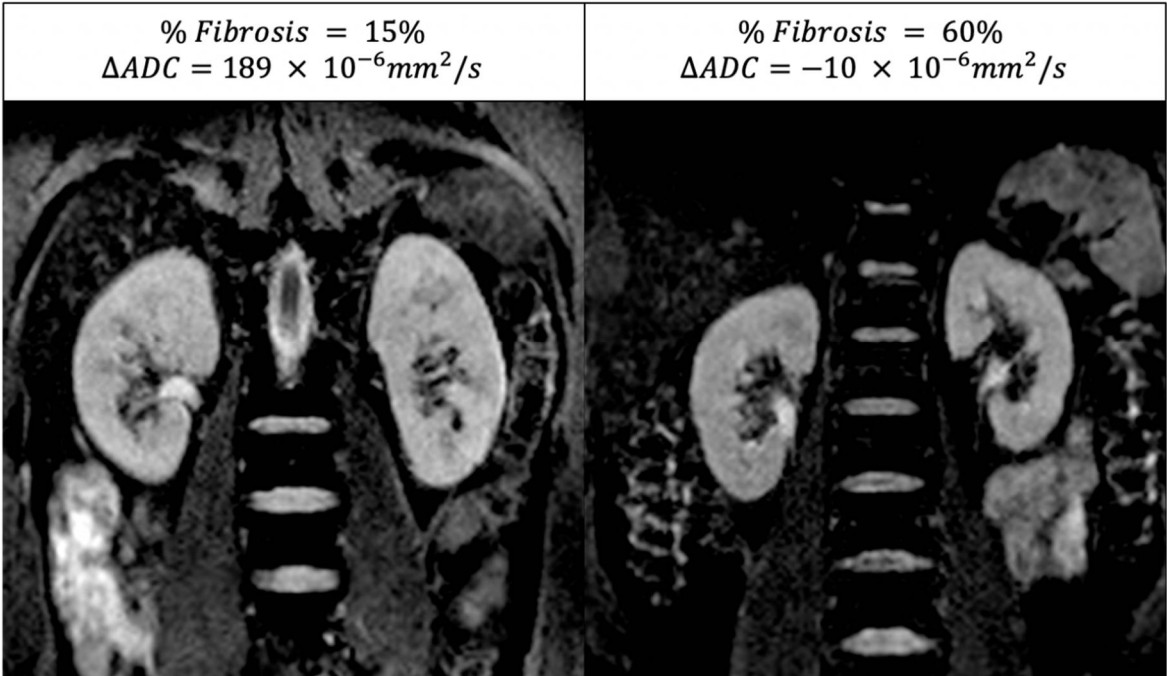

**Figure 3.** Example of an ADC map in two diabetic patients. On the left, the cortical ADC is higher (brighter) than the medullary ADC (positive $\Delta ADC = 189 \times 10^{-6}$ mm$^2$/s) in a patient with a small amount of fibrosis (i.e., 15%) on the renal biopsy. On the right, the cortico-medullary ADC difference is lost (negative $\Delta ADC = -10 \times 10^{-6}$ mm$^2$/s) in this diabetic patient with a higher amount of renal fibrosis on the renal biopsy (60%).

### 3. Renal MRI and Diabetic Kidney Disease

The application of MRI to kidney diseases is rapidly increasing, but most studies included CKD patients of different aetiologies, and few included exclusively DKD patients in their study populations. Most studies used BOLD-MRI and DWI-MRI, whereas much fewer data are available on T1 and T2 mapping, PC- and ASL-MRI. Here, we provide a non-exhaustive overview of recent studies in patients with CKD or DKD. No specific search strategy was applied, but the authors chose the studies based on their presumed clinical relevance and expected or demonstrated impact in this relatively new research area.

### 3.1. T1 and T2 Mapping

The number of studies that have applied T1 or T2 mapping to patients with DKD is limited. Most studies focused on CKD, irrespective of its cause. In one of the largest studies, Wu et al. reported higher cortical T1 values in 119 CKD patients with glomerulonephritis than in the controls; the T1 values correlated positively with creatinine and cystatin C, and negatively with eGFR and kidney length [30]. Further, T1 correlated strongly with the degree of fibrosis in the 43 patients who underwent a kidney biopsy. However, patients with DKD were not included in this study. A recent study by Dekkers et al. did not find differences in cortical nor medullary T1 values between 15 healthy volunteers and 11 patients with DKD, but the cortico-medullary ratio of T1 was significantly higher in patients with DKD [31]. Whether T1 or T2 values or their cortico-medullary ratio predict outcomes, or can be used to differentiate DKD from other causes of CKD, is currently unknown and the subject of several ongoing studies.

### 3.2. PC-MRI

Only limited studies using renal PC-MRI in diabetic patients are available. In a renal blood flow validation study in 25 patients with type 2 diabetes (36% female), a good agreement between ASL, delayed contrast enhancement (DCE), and PC RBF was observed on average, but not in individual patients [32]. Of interest, PC-MRI showed a significantly smaller reproducibility error than ASL [32]. PC-MRI has mainly been used in patients with suspected or confirmed renal artery stenosis, and fewer studies have focused on CKD or DKD. In the largest study to date by Khatir et al., 62 CKD patients (23% with DKD) and 24 age- and sex-matched controls underwent PC-MRI, BOLD-MRI, and chromium 51-labeled EDTA to measure GFR. Single-kidney RBF was 28% lower in CKD patients in comparison with controls (319 vs. 443 mL/min, $p < 0.001$), whereas measured GFR was 73% lower (36 vs. 97 mL/min) [33]. Separate analyses of DKD patients were not shown. The finding that the GFR decreased more than RBF was explained by the authors as a way for the kidneys to preserve kidney oxygenation despite reduced blood flow. Indeed, the filtration fraction of CKD patients was lower, and the R2* values did not differ between the CKD patients and the controls. In a randomised, double-blind, placebo-controlled, crossover trial in adults with type 1 diabetes and albuminuria, a single 50 mg dose of the SGLT2 inhibitor dapagliflozin and placebo in random order, separated by a two-week washout period, did not change renal perfusion or blood flow, but improved renal oxygenation [32,34].

### 3.3. ASL-MRI

As outlined in the introduction, vascular rarefaction plays an important role in the development and progression of DKD. In addition, the angiotensin II-induced vasoconstriction of efferent arterioles leads to reduced perfusion of the vasa recta, which in turn predisposes to renal hypoxia. The assessment of renal micro-circulation with ASL-MRI has demonstrated a reduction in cortical perfusion of 28% in patients with DM compared to healthy controls [35]. In a prospective study by Prasad et al. that included 54 individuals (41 with DKD, 13 controls), ASL-MRI-assessed cortical perfusion was an independent predictor of annual eGFR decline [36]. These data are encouraging, but larger studies including hard renal outcome are needed (and ongoing) to confirm the potential of ASL-MRI to predict renal outcomes.

### 3.4. BOLD-MRI

As sustained hyperglycaemia leads to mitochondrial dysfunction and inefficient electrolyte transport in the renal tubules, tissue hypoxia has historically been considered a central mechanism in the development of DKD. It is therefore no surprise that BOLD-MRI has been frequently applied to patients with DKD, albeit with conflicting results. In 2008, Inoue and colleagues reported that the cortical T2* values correlated with eGFR in 76 CKD patients without diabetes, but not in 43 patients with DKD [37]. However, they performed BOLD-MRI at 1.5T, and the control group was too small to assess whether hypoxia was in-

deed present in patients with DKD. A more recent study performed in China at 3T reported that the medullary R2* values were higher in 30 DKD patients versus 15 controls [38], but another study (also performed at 3T, in individuals of European descent) could not confirm these results, and found no differences in renal R2* between 18 patients with early DKD (stage 1–3a) and 18 age- and sex-matched controls [39]. The reasons for these conflicting results remain unclear. This could be due to ethnical differences in oxygenation, or also due to differences in the circulating glucose levels at the moment of the scan [40]. Moreover, all the abovementioned studies include a relatively small number of individuals, and larger studies are once more needed to draw definitive conclusions.

Nevertheless, prospective studies have demonstrated that BOLD-MRI can predict consecutive decline in eGFR. In a study by Sugiyama et al. of 91 Asian CKD patients (41.8% with DM), T2*, and the urine protein-to-creatinine ratio were the only independent predictors of eGFR decline [41]. In the abovementioned study by Prasad et al., the cortical R2* values were higher in DKD patients than in the controls, whereas the furosemide-induced change in R2* was inversely associated with annual eGFR decline [42]. Finally, in a Swiss cohort of 112 CKD patients (22% with DKD) and 71 controls, CKD patients with high cortical R2* values (>90th percentile) had a faster yearly eGFR decline and needed kidney replacement therapy three times more frequently than those with lower R2* values [43]. These studies illustrate the potential of BOLD-MRI, alone or in combination with other techniques, to be of benefit to patients with DKD.

*3.5. DWI-MRI*

Several studies have demonstrated the utility of DWI-MRI in DKD. Lu et al. were among the first in 2011 to notice that ADC was lower in T2D patients than in controls [44]. In the same year, Inoue et al. compared 50 diabetic people with different levels of eGFR and albuminuria to 76 healthy controls and found that the cortical ADC was lower in patients with DKD, and correlated linearly with the eGFR: the lower the eGFR, the lower the ADC [37] More recently, Cakmak et al. [45] confirmed that ADC correlated strongly with the clinical categories of DKD in 78 T2D individuals with DKD. A substantial association between ADC and eGFR has also been seen in several other studies [46,47]. Moreover, diabetic murine models demonstrated a direct relationship between ADC and interstitial fibrosis [48,49]. A recent study in CKD patients who underwent renal biopsy of either the native kidney (22% of whom had DKD) or the allograft kidney (none with DKD) confirmed this relationship in humans, and reported that the corticomedullary ADC difference (ΔADC) correlates better with interstitial fibrosis than absolute corticomedullary or medullary ADC values [50]. As a result, patients with interstitial fibrosis might be identified more accurately by ΔADC than by the individual cortical or medullary ADC values. However, the number of diabetic patients who underwent simultaneously renal MRI and kidney biopsy was limited, and confirmation in larger studies such as the iBeat project (see below) is needed.

A recent breakthrough study in 197 patients (154 transplanted and 43 CKD patients) by Berchtold et al. [51] has shown that ΔADC is an independent predictor of the composite endpoint decline in eGFR of ≥30% and/or dialysis initiation; those with a negative ΔADC had a more than five-fold higher risk of reaching the composite endpoint. However, only 6.1% of the patients had diabetes in this study. Once more, although these studies show that DWI-MRI can be of high interest to patients with DKD, more studies that focus exclusively on this patient group are needed.

## 4. Multiparametric MRI

In recent years, it became possible to perform different MRI techniques within one MRI session of a duration of ~1 h [52]. So far, only a limited number of centres have integrated this so-called multiparametric MRI in their research protocols, but technically, this technique can be performed on any modern MRI scanner once the sequences are installed. When the expertise is locally not available, specialised companies offer such services and have qualified teams to analyse the acquired images.

Makvandi et al. [53] recently performed a multiparametric MRI (mpMRI) study in patients with diabetic kidney disease. In this cross-sectional study, 20 healthy volunteers and 38 subjects with T2D and DKD with an eGFR of 15–60 mL/min/1.73 m$^2$ underwent mpMRI. The mpMRI protocol included PC, ASL, BOLD, T1 mapping, and DWI, and took on average of 50 min. They observed that anatomical T1-weighted MRI indicated high corticomedullary contrast in healthy volunteers, decreased contrast in stage G3 patients, and essentially no contrast in stage G4–G5 individuals. The MRI biomarkers of kidney microstructures, i.e., ADC of the cortex and medulla, and oxygenation in the medulla measured by BOLD-R2*, could also distinguish healthy volunteers from DKD patients, but PC-assessed renal blood flow (RBF) and ASL-assessed perfusion of the cortex best differentiated healthy volunteers from DKD patients.

Inoue and colleagues carried out a prospective mpMRI study in 151 CKD patients (29 with DKD) and compared the MRI results to the eGFR decline over time. All patients underwent BOLD-, ASL-, and DWI-MRI, as well as T1-mapping on a 3.0T scanner at baseline. The entire acquisition took roughly 30 min. The patients were followed for an average of 3.75 years, and eGFR slopes were calculated based on the creatinine values of at least three follow-up visits. In multiple regression analysis adjusted for several covariates including baseline eGFR and the degree of proteinuria, only BOLD-T2* was a predictor of eGFR decline [54]. However, the number of patients with DKD was too low to draw conclusions for this patient category. The ongoing European iBeat study performs a similar mpMRI protocol, and aims at recruiting 500 type 2 DM patients with eGFR $\geq$ 30 mL/min/1.73 m$^2$. Therefore, this study will probably provide a definite answer to the question of whether renal MRI can predict eGFR and kidney failure in patients with DKD [55].

## 5. Perspectives and Conclusions

Renal MRI has moved a long way from imaging strictly anatomical details, and can now provide information on renal function, perfusion, metabolism, and microstructure. Many of the techniques have been validated against gold standards, and international efforts have led to a more standardised approach in patient preparation, image acquisition, and analysis. Although several studies have reported the potential of renal MRI to predict renal function decline, only a few have included hard endpoints such as the doubling of serum creatinine or the need for kidney replacement therapy. Importantly, diabetic kidney disease has only been studied within larger study populations that group together different aetiologies of CKD. Although DKD has many features in common with other forms of CKD (such as vascular rarefaction and interstitial fibrosis), there are also differences. For example, the BOLD signal is influenced by circulating glucose levels [40]. In addition, the amount of renal sinus fat is significantly larger in patients with diabetes, correlates positively with HbA1c, and may influence other MRI parameters [56].

Several large studies that only include DKD patients are actually underway (notably iBeat) [55], and will provide essential data considering the particularities of DKD patients and the predictive power of renal MRI in diabetic patients.

Another often-mentioned hurdle to the introduction of renal MRI into clinical practice is its cost in comparison to urine or blood biomarkers. These costs not only include scanning time, but also the time-consuming analysis of the images by a specialist. Efforts are under way to automate the analysis process (e.g., machine learning approaches), but close collaboration with the vendors will be essential to integrate these analyses programs in scanners, and make the use of MRI in daily practice more user-friendly and attractive for clinicians worldwide.

Although MRI exams remain expensive, their costs are decreasing thanks to reductions in scanning times and larger availability due to an increasing number of MRI centres. The costs may partially be offset if renal MRI can replace renal ultrasound in some cases. Renal ultrasound is universally the first-line radiological exam performed in the work-up of CKD and DKD patients. As renal MRI can provide more high-quality anatomic and possibly

prognostic information in one session, and is less prone to artefacts in obese persons, its wider use may obviate the need for renal ultrasound. Notwithstanding these advantages, future studies should demonstrate that renal MRI is also viable from a health economic point of view. In our opinion, this is only possible if a baseline renal MRI allows the successful stratification of ambulatory DKD patients into "fast progressors" and "non-progressors", and does better than current methods such as eGFR, proteinuria, and the Kidney Failure Risk Equation Score. Progressors would benefit from more regular follow up, whereas non-progressors would be seen less often than is the case in standard care (thereby reducing costs elsewhere in the healthcare system). This hypothesis should be tested in a prospective randomised controlled study that compares the costs and outcomes of such an 'MRI-based' approach with standard care.

Renal MRI has the potential to replace kidney biopsies in some, but certainly not all situations. For the correct diagnosis and the exclusion of concomitant glomerulonephritis, kidney biopsies remain essential. However, when the main question of the clinician is, for example, to assess the degree of fibrosis, renal MRI will be an interesting option, in analogy with elastography techniques applied to the liver. This may allow non-invasive character-isation of DKD into those with different pathophysiology (classic diabetic nephropathy versus ischaemic or fibrotic processes, for example) that in the future may help guide more personalised therapy decisions.

Renal MRI also has huge potential in drug research, as it can provide a comprehensive overview of the renal haemodynamic and metabolic effects of a drug, minutes or several hours after its administration. In fact, renal MRI has already been used in several studies to clarify the mechanism of action, for example of SGLT2 inhibitors [34], and more studies are ongoing.

For all of these reasons, we believe that renal MRI will play an important role in the management of patients with diabetic and non-diabetic kidney disease in the future. The results of ongoing and future large-scale studies will tell us whether this is merely wishful thinking or reality, but we have reasons to be optimistic.

**Author Contributions:** All authors contributed to the conception of the work and substantively revised it. M.P. wrote the original draft, and I.A. wrote the paragraphs on DWI MRI. All authors have read and agreed to the published version of the manuscript.

**Funding:** This work is partially supported by the Swiss National Foundation (IZCOZO_177140/1).

**Institutional Review Board Statement:** Not applicable.

**Informed Consent Statement:** Not applicable.

**Data Availability Statement:** Not applicable.

**Conflicts of Interest:** M.P. declares consultancy activities for Antaros Medical and Novo Nordisk.

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
