# Peer review of "Magnetic Resonance Imaging to Diagnose and Predict the Outcome of Diabetic Kidney Disease—Where Do We Stand?"

_kidneydial, doi:10.3390/kidneydial2030036_

Round 1
Reviewer 1 Report
The review by Pruijm et al. is a timely and important paper, reviewing the applications of multi parametric MRI in DKD patients.
I really enjoyed reading the paper. I am however a bit concerned that the initial parts on the individual sequence etc. might be a bit too difficult to follow for people not doing MRI themselves.
Its difficult to explain everything in only a few sentences and maybe it would be worthwhile to move these descriptions to an appendix? I would think that most of the readers would be more interested in the second part, which is on what it means for DKD and how it can be used.
Author Response
The review by Pruijm et al. is a timely and important paper, reviewing the applications of multi parametric MRI in DKD patients.
I really enjoyed reading the paper. I am however a bit concerned that the initial parts on the individual sequence etc. might be a bit too difficult to follow for people not doing MRI themselves.
Its difficult to explain everything in only a few sentences and maybe it would be worthwhile to move these descriptions to an appendix? I would think that most of the readers would be more interested in the second part, which is on what it means for DKD and how it can be used.
We agree that the first part of the previous version focused a lot on the technical background of the different MRI techniques. We have followed the recommendation of the reviewer and simplified as much as possible this sections. We now refer to other publications, especially those published by the COST action group for more technical details. As a consequence, the first section has been shortened.
Reviewer 2 Report
the paper focuses on MRI results as possible predictors of renal funcional inpairment, particularly in diabetes type 2 patients.
Te AA refer to recently published consensus based recommendations formultated by international experts as part of the COST action CA16103 PARENCHIMA (see https://www.renalmri.org).
I think that, while the recommendations are an original contribution to clinical practice, the submitted paper is less intriguiging, as the aim is to discuss the technical principle underlying each tecnique.
I have some concerns about methods, as they have not been explicitated: databases, search string, screening for relevance, quality assessment, list of included and excluded studies...are not described, so that i am not certain that results are complete and unbiased
Author Response
the paper focuses on MRI results as possible predictors of renal funcional inpairment, particularly in diabetes type 2 patients.
Te AA refer to recently published consensus based recommendations formultated by international experts as part of the COST action CA16103 PARENCHIMA (see https://www.renalmri.org).
I think that, while the recommendations are an original contribution to clinical practice, the submitted paper is less intriguiging, as the aim is to discuss the technical principle underlying each tecnique.
I have some concerns about methods, as they have not been explicitated: databases, search string, screening for relevance, quality assessment, list of included and excluded studies...are not described, so that i am not certain that results are complete and unbiased.
We thank the reviewer for this pertinent comment. The first part of the previous version of the article focused indeed a lot on the technical principles of each technique, and may have given the impression that our only aim was to familiarize the reader with the technical basics of MRI. However, we also wanted to focus on the role that renal MRI can and may play in the future clinical care of patients with diabetic kidney disease (DKD). In the revised version, we have therefore reduced the technical part, and explain our aims more explicitly in the introduction. This review intentionally includes the personal opinions of the authors, and is therefore not without bias. This is also true for the literature search.
Concerning the methods, we agree that we did not perform a systematic review based on pre-defined search terms and quality criteria. Instead, we wanted to provide a short summary of the studies that will have-according to our opinion- the highest impact in this field. Once more, we wished to transmit our personal view, and we therefore preferred to select a limited number of papers, instead of performing a systematic review as some of the authors had done for the statement paper published in the NDT Supplement in 2018. In the revised version, we underline in more detail the aims and possible shortcomings of this review.
In answer to the comment that our paper is less intriguing than the consensus based recommendation papers, we wish to mention that the previously published statement papers focused either on the technical recommendations of renal MRI (in MAGMA, 2020), or on a comprehensive overview of all the studies performed in humans in each MRI technique at that time point, in patients with or without CKD (NDT Supplement 2018). The current review focuses on patients with diabetic kidney disease, and targets a different audience. We hope that our article will not only be read by nephrologists, but also by other physicians in charge of patients with diabetes, including internists, endocrinologists, diabetologists and cardiologists. We believe that the actual structure of the article (a brief technical overview, followed, by a short summary of clinical studies and our vision for the future) represents the best way to reach this target audience. We have underlined the target audience and aims in more detail in the revised version, and state that this is an opinion-based review, albeit written by experts in the field.
Reviewer 3 Report
Well written and comprehensive review, illustrations of high quality and informative. Except for minor improvements in style and language - no other criticism.
Author Response
Well written and comprehensive review, illustrations of high quality and informative. Except for minor improvements in style and language - no other criticism.
We thank the reviewer for this positive feedback. We have revised the paper for style and language, we hope it’s OK now.
Round 2
Reviewer 2 Report
The method section is still weak, there are no informations about the searche strategy and the criteria for incuding papers